

# Recent advances and challenges on application of tissue engineering for treatment of congenital heart disease

Antonia Mantakaki[1,*], Adegbenro Omotuyi John Fakoya[2,*] and Fatemeh Sharifpanah[3]

[1] Surgical Division, Teleflex Incorporated, Bristol, United Kingdom
[2] Department of Anatomical Sciences, University of Medicine and Health Sciences, Basseterre, St. Kitts and Nevis
[3] Department of Physiology, Faculty of Medicine, Justus Liebig University, Giessen, Germany
[*] These authors contributed equally to this work.

## ABSTRACT

Congenital heart disease (CHD) affects a considerable number of children and adults worldwide. This implicates not only developmental disorders, high mortality, and reduced quality of life but also, high costs for the healthcare systems. CHD refers to a variety of heart and vascular malformations which could be very challenging to reconstruct the malformed region surgically, especially when the patient is an infant or a child. Advanced technology and research have offered a better mechanistic insight on the impact of CHD in the heart and vascular system of infants, children, and adults and identified potential therapeutic solutions. Many artificial materials and devices have been used for cardiovascular surgery. Surgeons and the medical industry created and evolved the ball valves to the carbon-based leaflet valves and introduced bioprosthesis as an alternative. However, with research further progressing, contracting tissue has been developed in laboratories and tissue engineering (TE) could represent a revolutionary answer for CHD surgery. Development of engineered tissue for cardiac and aortic reconstruction for developing bodies of infants and children can be very challenging. Nevertheless, using acellular scaffolds, allograft, xenografts, and autografts is already very common. Seeding of cells on surface and within scaffold is a key challenging factor for use of the above. The use of different types of stem cells has been investigated and proven to be suitable for tissue engineering. They are the most promising source of cells for heart reconstruction in a developing body, even for adults. Some stem cell types are more effective than others, with some disadvantages which may be eliminated in the future.

Corresponding author
Adegbenro Omotuyi John Fakoya,
gbenrofakoya@gmail.com

# INTRODUCTION

Congenital heart diseases (CHD) refer to the abnormal formation of the infant's heart, great thoracic vessels and heart valves during intra-uterine development (*National Congenital Heart Disease Audit Report 2012–2015, 2016*). CHD is different from the acquired heart diseases which occur because of lifestyle or aging (*British Heart Foundation, 2016*). The

abnormalities are structural defects, such as valve defects, intravascular or intracardial stenosis, congenital arrhythmias or cardiomyopathies which greatly affect the early and future life of a CHD patient (*National Congenital Heart Disease Audit Report 2012–2015, 2016*; *Okudo & Benson, 2001*). People born with CHD need immediate medical care after birth which further continues throughout their lives. In 2010, it was estimated that only in the USA approximately 2.4 million people suffered from CHD and more than half of them were adults (*Gilboa et al., 2016*). In Europe, for instance, for the period of 2000 to 2005, about 36,000 live births per year were diagnosed with CHD (*Dolk, Loane & Garne, 2011*). The number grows bigger when including the unborn children who were diagnosed with CHD and died either by termination of pregnancy or by intrauterine death or even neonate death (*Dolk, Loane & Garne, 2011*). In the UK, about 8 in every 1,000 live babies born have a heart or circulatory condition (*National Congenital Heart Disease Audit Report 2012–2015, 2016*). Some estimate those numbers to be higher and, commonly, percentages of each type of CHD change depend on the geographical area of investigation (*Hoffman & Kaplan, 2002*; *Van der Linde et al., 2011*), Table 1 further mentions other CHD frequencies for other countries. CHDs not only have an effect on the individual's and their family's lives but also have a huge financial impact on the healthcare system. According to NHS England for the financial year 2013/14 the total spent on CHD was £175 million (*NHS England, 2015*) and in the US the total cost for CHD treatment in 2008 was approximately $298 billion (*Lundberg, 2013*). In general, the number of children and adults being diagnosed with CHD increases due to the improved technology of diagnostic tools (*Hoffman & Kaplan, 2002*).

CHD can be diagnosed using transabdominal fetal Doppler echocardiography. Such prognostic protocols are performed in high-risk groups of pregnant women, like those with a family history of CHD (*Nayak et al., 2016*). In adults with CHD, the most effective diagnostic practice is transesophageal echocardiography, electrocardiogram, pulse oximetry, X-rays, cardiac catheterization and MRI (*Sun et al., 2015*). The CHDs are managed by surgery, and the efficiency of this approach is largely dependent on the materials which are used during the surgery. These materials are expected to be close to the native cardiac tissue in both structure and function. In structures, CHD could present extremely complicated malformations which cannot be spontaneously or by singular surgical procedure reconstructed, hence the dire need for more research into biomaterials for Tissue Engineering (TE). The recent extensive research focuses on possible ways to fabricate a near ideal tissue. So far, TE appears to be the way forward in creating ideal tissue that can probably mimic the native heart tissue both in structure and function. TE refers to creation of functional three-dimensional tissue using biomaterials and cells for replacement or restoration of damaged organs and/or parts of them. TE is the most promising approach at the present for CHD, as treatment can be "patient-specific" and the engineered tissue could adjust to the developing body of the recipient. Many would think that TE is an idea which conceived and developed in a very recent past. However, it has been proven that tissue regeneration and TE is a concept which was born thousands of years ago, and it has inspired Greek mythology, history, arts, and religion. In art, religion inspired the well-known painting of "Healing of Justinian" based on the miracle of St.

**Table 1  Frequencies of CHDs in some regions.**

| | |
|---|---|
| United States | Affects 1% of live births (*Krasuski & Bashore, 2016*) |
| South America | Colombia: 1.2 per 1,000 live births<br>Brazil (Minas Gerais): 9.58 in 1,000 live births<br>Brazil (Londrina): 5.49 in 1,000 live births (*Pedra et al., 2009*) |
| Mexico | Affects 6–8 per 1,000 newborns. Drawing to the conclusion that there about 12,000 or 16,000 babies living with CHD (*Calderón-Colmenero et al., 2013*) |
| Asia | Affects 9.3 per 1,000 live births (*Van der Linde et al., 2011*) |
| Europe | Affects 8.2 per 100 live births (*Van der Linde et al., 2011*) |
| United kingdom | Affects about 9 in every 1,000 babies (*NHS, 2018*) |
| Russia | Affects 2.7–3.8 per 1,000 newborns estimating as 86 newborns per year being affected with CHD (*Postoev, Talykova & Vaktskjold, 2014*) |
| Australia | Affects 8–10 cases per 1,000 live births. Resulting in 2,400–3,000 newborns with CHD each year. About 65,000 adults are living with CHD (*HeartKids, 2018*) |
| Africa | Mozambique: 2.3 in 1,000 live births<br>Northern Nigeria: 9.3% (122 of 1,312 patients) (*Zühlke, Mirabel & Marijon, 2013*) |
| Canada | Affects 1 in 80–100 live births (*Canadian Heart Alliance, 2018*) |

Cosmas and St. Damian, physicians and Christian martyrs who appear to have transplanted the leg of an Ethiopian to the body of a patient (*Durant, 2018*). The closest to an artificial replacement of a body part was discovered in Egypt on a mummy which had a wooden replacement of the hallux (*Finch, 2011*). However, today, TE involves a combination of creating scaffolds and cell seeding. With regards to the heart, the most commonly used and known artificial parts are the mechanical heart valves and conduits (*Zilla et al., 2008*; *Gott, Alejo & Cameron, 2003*). The first artificial heart valve was placed on live patients only in the last century (*Zilla et al., 2008*; *Perry et al., 2003*). In contrast to the adult heart, infants' and children's hearts regenerate in a larger capacity because the regenerative ability is proportionally correlated to age (*Rupp & Schranz, 2015*). Additionally, there is an insufficient number of heart donors which becomes more challenging because of the heterogeneous relation of recipient-donor and the diverse range of CHD. These points result in high mortality rates and further financial costs to the healthcare systems (*Saxena, 2010*). The mortality rate among all patients who are waiting for any type of organ transplantation is highest in infants who wait for heart transplant (*Dodson, 2014*; *Homann et al., 2000*). In 2003, some evidence was presented to support the regenerative ability of the adult heart (*Beltrami et al., 2003*). This evidence shows the existence of cardiac stem/progenitor cells which can differentiate into new cardiomyocytes and participate in cardiac regeneration (*Beltrami et al., 2003*). However, only a small number of preclinical studies have focused on CHD treatments (*Ebert et al., 2015*; *Tarui, Sano & Oh, 2014*). Stem cells (SC) have been widely investigated mainly for myocardial infarction (MI), as it is currently the leading cause of morbidity and mortality worldwide (*Fakoya, 2017*). Cell

seeding is a fundamental component of TE. Several studies have examined the possibility of direct cell delivery in the damaged area, cardiac patch implantation and engineered heart tissue, with the former being the most popular (*Ebert et al., 2015*; *Fakoya, 2017*; *Feric & Radisic, 2016*). All possible types of stem cells are under investigation to identify the most appropriate cell types for tissue engineering using in corrective surgery of CHD. This review looks into the congenital heart diseases, biomaterials and scaffolds, and, types of stem cells used in TE.

## METHOD

This paper was based on review articles and reports in reputable peer-reviewed journals and government websites. The research was conducted using Medline on OvidSP, PubMed, google scholar, website, books, e- books, and reports. The words "congenital heart disease", "tissue engineering", "surgical treatment", "stem cells", "scaffolds", "biomaterials" and a combination of those were used to retrieve literature from the databases.

## CONGENITAL HEART DISEASE: TYPES, MALFORMATIONS, PRESENTATIONS AND INTERVENTIONS

CHD includes a diverse range of conditions which shows a variety of symptoms, indications, and malformations detected during pregnancy or after birth (*Sun et al., 2015*). However, these malformations are much influenced by the age of diagnosis (*Hoffman & Kaplan, 2002*). The etiology of CHD is unknown, but it is generally accepted that many factors or a combination of them could contribute to CHD and considered to be caused by multifactorial inheritance. These factors could be genetic, epigenetic or environmental factors such as alcohol and drugs consumption, as well as viral infections like Rubella (*Sun et al., 2015*). The severity of the disease varies, and a number of malformations could be present in each case. Based on the severity of CHD, they are categorized to mild, moderate, and severe CHDs, which the latter is subcategorized to Cyanotic and Acyanotic lesions (*Saxena, 2010*). The most frequent type of severe CHD is Ventricular Septal Defect (VSD) (*Penny, 2011*). VSD could cause myocardial defects which disappear in the first year of the infant's life (*Penny, 2011*). Nevertheless, the VSD could also cause some malformations which can be managed only by surgical intervention, that is, infant pulmonary hypertension (*Hoffman & Kaplan, 2002*; *Penny, 2011*). The other CHD type is Atrial Septal Defect (ASD) which is usually asymptomatic and in most of the case will only be diagnosed in adulthood (*Hoffman & Kaplan, 2002*). Atrioventricular septal defects (AVSD) is mainly observed in trisomy 21 (*Hoffman & Kaplan, 2002*). AVSD is usually characterized by "complete AV-canals with one common AV valve for both ventricles and an interatrial and intraventricular communication" and requires surgical correction. The results of long-term patient follow up after operation have shown very satisfactory survival rate (*Boening et al., 2002*). Another type of CHD, tetralogy of Fallot (ToF), is characterized by VSD, pulmonary stenosis, right ventricular hypertrophy and over-riding of the aorta (*Apitz, Webb & Redington, 2009*). Infants who suffer from ToF will require immediate surgical intervention for better survival rates and avoid cyanosis, a result of

inadequate pulmonary blood flow (*Apitz, Webb & Redington, 2009*). Calcific Aortic Valve (CAV), another type of CHD, is a disease which progresses slowly and results to a mild valve thickening and obstructing blood flow, aortic sclerosis or severe calcification with impaired leaflet motion (*Freeman & Otto, 2005*). CAV presents many similarities with arteriosclerosis in adults which is caused by lifestyle or aging (*Freeman & Otto, 2005*). However, CAV is a congenital, progressive disease which could be diagnosed in patients less than one year of age, and those in childhood or even adulthood (*Smith et al., 2015*; *Medscape, 2015*). Table 1 summarizes the frequencies of congenital heart diseases, while Table 2 summarizes the types of CHDs, presentations and possible management.

Some types of the CHDs are misdiagnosed, undiagnosed, or diagnosed very late in life which could make a successful treatment challenging (*Torok, 2015*).

CHD patients usually require medication, cardiac catheterization or a series of surgical interventions throughout their lives (*Sun et al., 2015*; *Homann et al., 2000*). This, among other risks, increases the chances of HLA-sensitization which eventually makes it more difficult to find a cross-match (*Homann et al., 2000*). The main challenge with artificial materials is the alterations in size and function of the heart from the neonatal period, infanthood, and to adulthood, to which current artificial materials cannot adjust to (*Sun et al., 2015*). The need for early intervention is essential for normal physical and cognitive development (*Torok, 2015*).

The use of bioprosthetics including allografts and xenografts is widely used for treatment of CHD as they present good survival rates in the patients and fewer interventions throughout their lifetimes (*Dodson, 2014*). The observed symptoms, defects, and complications upon bioprosthetic treatment of CHD patients can differ from case to case (*Penny, 2011*; *Torok, 2015*).

## Heart, a complex organ for tissue engineering

Unlike other human tissue, cardiac tissue is a more complex tissue considering not only because of its mechanical and structural function but also due to its electrical properties as well. The human heart mainly consists of cardiomyocytes, functions as a blood pump which is regulated by the electrical signal generated by the pacemaker cells in the sinoatrial nodes. This signal is directed and spread through the atrioventricular node to Purkinje fibers, and this is highly important for the direction of blood flow (*Files & Boucek, 2012*). The diastolic and systolic function of the heart is necessary to be synchronized and adjusted according to the body's needs. The isolation of this signal from the rest of cardiac tissue is as important as this electrical signal by itself. This importance is achieved by the extracellular matrix (ECM) of heart which is also responsible for mechanical support and endurance (*Files & Boucek, 2012*). Aside from the functional aspects briefly mentioned above, the structural aspects (the cardiomyocyte and its ECM) are other factors for strong consideration in the success of TE. To yield an engineered tissue with the best functionality, this engineered tissue must be similar in every sense to the native tissue. The ECM of the heart is mainly a complex mesh of structural elements such as cardio fibroblasts and collagen fibrils, and non-structural elements such as proteoglycans, glycosaminoglycans, and glycoproteins (*Kaiser & Coulombe, 2016*; *Rienks et al., 2014*) among other components. Repairing the heart using

Mantakaki et al. (2018), *PeerJ*, DOI 10.7717/peerj.5805

**Table 2** Types, presentations and management of CHDs.

| Types of CHD | Presentations of CHD | Medical & surgical therapeutic approach to CHD |
|---|---|---|
| Atrial Septal Defects | ➢ This defect manifests as a hole in the wall (septum) that separates the chambers above (atria) from those below (ventricles)<br>➢ The volume of blood that flows through the lungs is increased over time due to the hole caused by the atrial defect resulting in damage to the blood vessels in the lungs<br>➢ Frequent respiratory or lung infections<br>➢ Difficulty breathing<br>➢ Tiring when feeding (infants)<br>➢ Shortness of breath when being active or exercising<br>➢ Skipped heartbeats or a sense of feeling the heartbeat<br>➢ A heart murmur, or a whooshing sound that can be heard with a stethoscope<br>➢ Stroke<br>➢ Swelling of legs, feet, or stomach area (*CDC, 2018a*). | ➢ **Medical monitoring:** the patient is monitored to see if the atrial septal defect would close on its own<br>➢ **Medications:** beta blockers (to maintain a regular heartbeat) or anticoagulants (to help reduce blood clots)<br>➢ **Surgery:** can be done through Cardiac catheterization or open-heart surgery<br>➢ **Follow-up care** (*CDC, 2018a*; *Mayo Clinic, 2018a*) |
| Hypoplastic Left Heart Syndrome | ➢ This defect affects the normal blood flow through the heart. The left side of the heart does not form correctly and as such it is considered a critical congenital heart defect<br>➢ The following structural malformations are observed in the left side of heart<br>a. The left ventricle is underdeveloped.<br>b. The mitral valves are not formed.<br>c. The aortic valve is not formed.<br>d. The ascending aorta is underdeveloped.<br>➢ The left side of the heart cannot pump oxygen-rich blood.<br>➢ Rapid, difficult breathing<br>➢ Pounding heart<br>➢ Weak pulse<br>➢ Poor feeding<br>➢ Being unusually drowsy or inactive<br>➢ Ashen or bluish color<br>➢ Dilated pupils<br>➢ Lackluster eyes that seem to stare (*CDC, 2018b*; *AHA, 2018*). | ➢ **Medication:** inpatient medications include prostaglandin E1, Dopamine and Potassium Chloride and outpatient medications are Furosemide, Digoxin and Captopril (*Patnana & Turner, 2018*).<br>➢ **Nutrition:** feeding tube or special high-calorie formula.<br>➢ **Surgery:**<br>a. Norwood Procedure: performed on the infant within 2 weeks of a baby's life.<br>b. Bi-directional Glenn Shunt Procedure: done on an infant around 4 to 6 months of age.<br>c. Fontan Procedure: performed on an infant around 18 months to 3 years of age (*CDC, 2018b*) |
| Tricuspid Atresia | ➢ Occurs in which the tricuspid valve is not formed leading to the underdevelopment of the right ventricle.<br>➢ The right side of the heart can't pump sufficient blood to the lungs.<br>➢ Problems breathing<br>➢ Ashen or bluish skin color (cyanosis)<br>➢ Poor feeding<br>➢ Extreme sleepiness<br>➢ Slow growth and poor weight gain<br>➢ Edema of the abdomen, legs, ankles and feet (*Mai et al., 2012*; *Mayo Clinic, 2018f*). | ➢ **Medications:** prostaglandins like Al-prostadil IV to keep open the ductus arteriosus.<br>➢ **Nutrition:** feeding tube<br>➢ **Surgery:**<br>a. Atrial Septostomy: performed in the first few days or weeks of a baby's life<br>b. Banding<br>c. Shunt Procedure: done within the first 2 weeks of a baby's life.<br>d. Bi-directional Glenn Procedure: performed around 4 to 6 months of the baby's life.<br>e. Fontan Procedure: done around 2 years of age (*Mai et al., 2012*). |

**Table 2** (*continued*)

| Types of CHD | Presentations of CHD | Medical & surgical therapeutic approach to CHD |
|---|---|---|
| **Tetraogy of Fallot** | ➤ Has a combination of four heart defects. This defect is a combination of pulmonary stenosis, ventricular septal defect, overriding aorta and right ventricular hypertrophy.<br>➤ Cyanosis<br>➤ Shortness of breath<br>➤ Rapid breathing especially during feeding or exercise<br>➤ Fainting<br>➤ Clubbing of fingers and toes<br>➤ Poor weight gain<br>➤ Fatigue during play or exercise<br>➤ Prolonged crying<br>➤ Irritability<br>➤ Heart murmur due to pulmonary stenosis (*Mayo Clinic, 2018e*; *Baffa, 2018*). | ➤ **Medication:** Prostaglandin E$_1$ infusion.<br>➤ **Surgery:**<br>a. Temporary surgery (palliative surgery): improve blood flow to the lungs.<br>b. Intra-cardiac repair: done during the first year after birth (*Mayo Clinic, 2018e*). |
| **Bicuspid Aortic Valve** | ➤ Has only two (bicuspid) cusps instead of three.<br>➤ A bicuspid aortic valve may result in the heart's aortic valve to narrow (aortic valve stenosis) which prevents the valve from opening completely, which reduces or blocks blood flow from the heart to the body.<br>➤ Trouble breathing<br>➤ Chest pain or pressure<br>➤ Fatigue<br>➤ Heart racing<br>➤ Light-headedness<br>➤ Fainting (*Northwestern Medicine, 2018*). | ➤ **Surgery:**<br>a. Aortic valve replacement<br>b. Balloon valvuloplasty<br>c. Aortic valve repair<br>d. Aortic root and ascending aorta surgery (*Mayo Clinic, 2018b*) |
| **Patent Ductus Arteriosus** | ➤ A persistent opening between the two major blood vessels leading from the heart.<br>➤ Large patent arteriosus can cause poorly oxygenated blood to flow in the wrong direction.<br>➤ Poor eating leads to poor growth.<br>➤ Sweating with crying or eating<br>➤ Persistent fast breathing or breathlessness<br>➤ Easy tiring<br>➤ Rapid heart taste (*Mayo Clinic, 2018c*) | ➤ **Medications:** NSAIDS (Advil, Infant's Motrin), or indomethacin (Indocin) (*Mayo Clinic, 2018c*)<br>➤ **Surgery:** Video-assisted thoracic surgical (VATS) repair<br>➤ **Catheter procedure:** Trans-catheter occlusion (*Cleveland Clinic, 2018*)<br>➤ **Watchful waiting** |
| **Pulmonic Valve Stenosis** | ➤ This defect affects the pulmonic valve in which a deformity on or near the valve causes it to be smaller and as such slows the blood flow.<br>➤ The narrowing is due to the underdevelopment of the valve during fetal growth. The cusps maybe defective or too thick or may not separate from each other well.<br>➤ Heart murmur<br>➤ Fatigue<br>➤ Shortness of breath, especially during exertion<br>➤ Chest pain<br>➤ Fainting (*Mayo Clinic, 2018d*) | ➤ **Surgery:**<br>a. Balloon valvulplasty<br>b. Open-heart surgery (*Mayo Clinic, 2018d*) |
**Table 2** (*continued*)

| Types of CHD | Presentations of CHD | Medical & surgical therapeutic approach to CHD |
|---|---|---|
| **Ventricular Septal Defect** | ➢ A fissure connecting the two ventricles of the heart. Size varies with each patient (*Spicer et al., 2014*)<br>➢ It can occur isolated or in association with other CHDs.<br>➢ There are three kinds: muscular, periventricular and supra-crystal. These are based on location within the septum (*Carminati et al., 2007*)<br>➢ **Blood shunting:** Depending on the size of the hole, blood flows from the left ventricle to the right.<br>➢ **Pulmonary hypertension**: The shunting of blood flow leads to increased ventricular output to the pulmonary artery. With time, this can lead to pulmonary hypertension.<br>➢ **Eisenmenger's Syndrome:** Rise in pulmonary vascular resistance leads to increase in right ventricular pressure. This can lead to reverse shunting of blood from the right to left. This leads to cyanosis (*Spicer et al., 2014*)<br>➢ Patients could also display clubbing (*Carminati et al., 2007*)<br>➢ Growth retardation: Increased blood flow to the lungs results in an increase in lung compliance. This increases the energy demand for respiration. Thus, an energy deficit is created where the infant does not consume as much calories as is burned. This impedes growth.<br>➢ **Airway Obstruction:**<br>a. Increased pulmonary blood flow increases the size of the pulmonary arteries. This can cause the physical obstruction of large and small airways.<br>b. There is also the possibility of the incidence of pulmonary edema due to increased blood flow. The combination of these events can lead to respiratory distress. Thus, symptoms such as wheezing, and tachypnea can be observed (*Spicer et al., 2014*).<br>c. Holosystolic/pansystolic murmur on auscultation (*Minette & Sahn, 2006*) | ➢ Smaller holes resolve themselves with time.<br>➢ **Surgery:** Usually done on larger fissures. It is indicated where patients express symptoms of heart failure, left heart overload and history of endocarditis.<br>➢ **Percutaneous techniques:**<br>a. These do not require opening the patient up.<br>b. Trans catheter approach: A catheter is threaded from an artery in the legs, or groin into the heart. A device is then placed to obstruct the hole in the ventricle (*Spicer et al., 2014*; *Carminati et al., 2007*). |
| **Total Anomalous Pulmonary Venous Condition** | ➢ The pulmonary veins are attached to the right atrium instead of the left.<br>➢ Usually associated with atrial septal defect.<br>➢ **Cyanosis:** Oxygenated blood from the lungs is pumped into the right atrium. It mixes with deoxygenated blood and passes through the atrial septal defect into the left atrium. This decreases oxygen supply to the body leading to cyanosis.<br>➢ **Pulmonary hypertension:** Some patients have constricted pulmonary veins that lead to pulmonary hypertension. This leads to pulmonary effusion.<br>➢ **Hypovolemia**: Some patients manifest with a narrow or restrictive atrial septal defect. This significantly reduces the blood flow to the body leading to hypovolemia (*NIH, 2018*). | ➢ **Surgery:** The pulmonary veins are surgically reattached to the left atrium.<br>➢ **Cardiac catheterization:** For the patients with a restricted atrial septal defect, a balloon pump is used to widen the fissure until corrective surgery can be carried out (*NIH, 2018*) |

Mantakaki et al. (2018), *PeerJ*, DOI 10.7717/peerj.5805

**Table 2** (*continued*)

| Types of CHD | Presentations of CHD | Medical & surgical therapeutic approach to CHD |
|---|---|---|
| **Trans-Position of the Great Arteries** | ➢ A condition whereby the aorta and the pulmonary arteries are transposed. The aorta arises from the right ventricle and leads to the lungs. The pulmonary artery arises from the left ventricle and leads to the body.<br>➢ It is comorbid with ventricular septal defect and patent ductus arteriosus.<br>➢ **Cyanosis:** Mixing of blood leads to supply of poorly oxygenated blood to the body (*Martins & Castela, 2018*) | ➢ **Medications:** Prostaglandin E1 is administered to keep the ductus arteriosus open.<br>➢ **Surgery:**<br>a. Balloon atrial septostomy: A catheter is threaded through the foramen ovale. A balloon is inflated to rip a fissure in the atrium (*Martins & Castela, 2018*) |
| **Truncus Arterious** | ➢ A condition where the truncus arteriosus of a fetus does not differentiate into an aorta and pulmonary vein. Thus, the patient only has one vessel exiting the heart<br>➢ **Cyanosis:** This leads to mixing of oxygenated and deoxygenated blood. Thus, the oxygen supply to the body is decreased<br>➢ **Congestive heart failure:** the excess of volume of blood flow to the heart increases pressure in the lungs. This would eventually lead to cardiac failure.<br>➢ Usually comorbid with ventricular septal defect (*Cincinnati Children Health, 2018*) | ➢ **Surgery:** The truncus arteriosus must be separated into two vessels. This would allow separate blood flow channels to the heart and body (*Cincinnati Children Health, 2018*) |
| **Ebstein's Anomaly** | ➢ A congenital malformation of the tricuspid valve<br>➢ The posterior and septal leaflets are displaced downwards. This leads to a downward enlargement of the right atrium.<br>➢ The walls of the right atrium become thin.<br>➢ It can be comorbid with patent foramen ovale or atrial septal defect.<br>➢ Patients can be asymptomatic but could also present with symptoms.<br>➢ **Cyanosis:** Shunting of the blood between patent foramen ovale and atrial septal defect leads to blood mixing between the left and right sides of the heart. This can lead to cyanosis.<br>➢ **Conduction irregularities:** Some patients present with arrhythmias (*Khan et al., 2018*) | ➢ **Surgery:**<br>a. Only required where patient manifests severe symptoms.<br>b. Cone procedure: Where the anterior septal leaflet is maneuvered and sewn to the true annulus. This attachment causes it to be conical in shape.<br>c. Valve replacement<br>d. Where the defective valves can be surgically replaced (*Khan et al., 2018*) |
| **Pulmonary Atresia** | ➢ Characterized by a restriction to blood flow from the right ventricle to the pulmonary artery. It could be due to malformation of the pulmonary valve or of the pulmonary artery itself.<br>➢ It can manifest with a ventricular septal defect where there are collateral arteries supplying the lungs.<br>➢ It could also manifest without a ventricular septal defect. Here the right ventricle is usually hypoplastic. It would usually be comorbid with a patent ductus arteriosus (*Safi, Liberthson & Bhatt, 2016*). | ➢ **Surgery:**<br>a. A shunt must be created between the pulmonary artery and the aorta. This can be done by administration of prostaglandin E to keep the ductus arteriosus open. It could also be doe surgically.<br>b. Fontan's procedure: Done for patients with a hypoplastic right ventricle. The right atrium is surgically connected to the pulmonary artery (*Safi, Liberthson & Bhatt, 2016*). |

Mantakaki et al. (2018), *PeerJ*, DOI 10.7717/peerj.5805

**Table 2** (*continued*)

| Types of CHD | Presentations of CHD | Medical & surgical therapeutic approach to CHD |
|---|---|---|
| **Aortic Stenosis** | ➢ Defect of the aortic valve that restricts its opening. <br> ➢ It can lead to ventricular hypertrophy which can eventually lead to heart failure. <br> ➢ There is also a possibility of development of atrial fibrillation. <br> ➢ Presence of systolic murmur (*Skybchyk & Melen, 2017*) | ➢ **Valve replacement:** There are two approaches: <br> a. Surgically <br> b. Trans-catheter approach (*Skybchyk & Melen, 2017*). |
| **Coarctation of the Aorta** | ➢ It is the constriction of the proximal end of the aorta leading to restriction to blood flow. <br> ➢ Patients can present with acidosis, cardiac failure, as well as shock after ductus arteriosus closes (*Johnson et al., 1998*) | ➢ **Surgery:** <br> a. Coarcted potion can be resected. And the ends of the artery can be re-anastomosed. <br> b. A patch can also be used to surgically dilate the artery. <br> ➢ **Balloon Angioplasty:** A catheter is threaded in to the aorta. A balloon is inflated to enlarge the aorta. |

materials which do not have or do not comply with the above characteristics and cannot work in harmony with the host-heart, will result in a non-efficient functioning heart accompanied by a series of complications. The concept and act of repairing the heart using various engineered techniques has evolved over the years from the use of artificial heart valves and grafts to bioprosthesis, and currently forward to the use of biomaterials and scaffolds cells.

## Artificial heart valve and grafts

Charles Hufnagel was the first to experiment on animals with an artificial valve which he designed in the 1940's (*Gott, Alejo & Cameron, 2003*). A few years later, the same type of valve was transplanted into humans (*Gott, Alejo & Cameron, 2003*). Nonetheless, Hufnagel's valve required changes which were succeeded by Harken-Soroff and later by Starr-Edwards ball valve (*Gott, Alejo & Cameron, 2003*). Following several improvements, the latest version is made of pure carbon as a lighter, smoother material for blood flow and more durable in comparison to other materials (*Zilla et al., 2008*). Due to the position of the valve, there is a high transvascular pressure which leads to 'impact wear' and 'friction wear' (*Zilla et al., 2008*). Further complications with the use of valve replacements include inflammation around the prosthesis and calcification of the valve itself (*Brown, 2005*). The main disadvantage of artificial valves is the thromboembolic risk which leads to lifetime treatment with anticoagulants. This type of treatment involves various complications and brings different risks to patient (*Zilla et al., 2008*). Similarly, to valves, artificial vascular grafts have been used for many years as a surgical treatment for CHD (*Chaikof, 2007*). Nevertheless, they provoke an inflammatory response, and they are much less flexible than the body's natural tissue (*Torok, 2015*). Originally porous fabric knitted of Dacron and polytetrafluoroethylene (PTFE) was used for stenosis treatment. Later alterations in the porosity of fabric were introduced to prevent the material's corruption (*Torok, 2015*). The first attempt to combine biomaterials with patient's cells was made by Wesolowki and colleagues by "preclotting the graft with the patient's blood" (*Torok, 2015*). However, future attempts to produce successful results created many doubts about the actual function of cells following citation on clot (*Torok, 2015*). Nonetheless, both artificial heart valves and conduits require continuous use of anticoagulants. Remarkably, they do not grow with the patients' heart as patients with CHD are highly likely to require surgical intervention when they are an infant or a child (*Torok, 2015*; *Ratner et al., 2004*).

## Bioprosthetics: allograft, xenografts, and autografts

Allograft heart valves and arterial grafts are collected from deceased humans. In contrast, xenografts are harvested from porcine and bovine animals including heart valves and carotid arteries (*Zilla et al., 2008*; *Perry et al., 2003*; *Brown, 2005*). Allografts and xenografts came into the picture as an alternative to artificial valves and conduits. Their main advantage is that they do not require lifetime treatment with anticoagulants (*Zilla et al., 2008*; *Hibino et al., 2010*; *Schmidt & Baier, 2010*). Animal tissue is treated with glutaraldehyde (*Hibino et al., 2010*). Glutaraldehyde is a five-carbon bifunctional aldehyde used to stabilize tissue to protect from chemical and enzymatic degradation and maintain "its mechanical integrity
and natural compliance" (*Hibino et al., 2010*). Also, treatment is necessary to reduce immunogenicity of the xenograft by decellularization and sterilization of the tissue (*Hibino et al., 2010*). Like xenograft, allografts also need treatment before transplantation and can even be cryopreserved (*Dodson, 2014*). Despite the great advantage, a number of complications are related to bioprosthetic grafts related to their preparation (*Hibino et al., 2010*).

The risk of cytotoxicity leading to inflammation, as well as the partial loss of mechanical properties of tissue, have been reported (*Hibino et al., 2010*). Moreover, calcification is often observed in infants and children with bioprosthetics. Many efforts are being made to find out an alternative treatment for heart bioprosthetics therapy. However, there is a controversy regarding their efficiency (*Hibino et al., 2010*).

In a study carried out by Homann and colleagues, the outcomes of 25 years using allografts and xenografts for reconstruction of the right ventricular outflow tract showed 66% survival at 10- years' follow up. Furthermore, patients with allografts had a mean reoperation-free interval time of 16 years in contrast to the xenograft recipients which this interval time is 10.3 years (*Dodson, 2014*). Allografts may present better outcomes, but they are not in abundance like xenografts. Therefore, many studies are concentrating on the development of tissue valves and vascular grafts created by stem cell-seeded on artificial or natural scaffold (*Perry et al., 2003*).

The relatively recent "CorMatrix" patch fabricated from the decellularized porcine small intestinal submucosa extracellular matrix (SIS-ECM) mainly composed of collagen, elastin, glycan, and glycoproteins have been introduced into cardiac surgery. SIS-ECM has not only been used in animal models for cardiac surgery (*Dodson, 2014*; *Homann et al., 2000*; *Files & Boucek, 2012*; *Kaiser & Coulombe, 2016*; *Rienks et al., 2014*; *Brown, 2005*; *Chaikof, 2007*; *Ratner et al., 2004*; *Hibino et al., 2010*; *Schmidt & Baier, 2010*; *Kurobe et al., 2012*; *Johnson et al., 1998*; *Ruiz et al., 2005*), but also in human studies, for cardiac and vascular reconstructions such as augmentation of the tricuspid valve (*Wainwright et al., 2012*), vascular repair of ascending aorta, aortic arch, right ventricular outflow tract, pulmonary artery, valvular reconstruction (*Scholl et al., 2010*), and closure of septal defects (*Quarti et al., 2011*). The study by *Witt et al. (2013)* reported a small risk of stenosis when SIS-ECM is used in the reconstruction of the outflow tracts and great vessels. Interestingly, the SIS-ECM has effectively proved the function in the high-pressure vessels (*Quarti et al., 2011*). The pitfall of this study, however, was the short follow up period.

A very common surgical practice for CHD is the use of pericardial patches for repairing the septal defect (*Torok, 2015*). The autologous pericardium is the best choice for infants as it is free, it does not provoke any immune-response, and it is sterile. Even though it requires some preparation before application, autologous pericardium creates less fibrotic tissue in comparison to Dacron (*Torok, 2015*). Allograft pericardium is available, but quite a few risk factors are associated with its use (*Torok, 2015*).

## Biomaterials and scaffolds for tissue engineering

Generally, scaffolds work as a primary base for cells to enhance and produce relevant tissue. The scaffolds should have specific morphological, functional, and mechanical properties

to support cells survival and differentiation (*Feric & Radisic, 2016*). Biomaterials used to produce scaffolds should be made of components which will accommodate the above characteristics and create a friendly cell microenvironment (*Feric & Radisic, 2016*). The previously mentioned characteristics apply for all different types of engineered tissue, and the goal is to mimic host tissue in the best possible way. With regards to artificial and bioprosthetic cardiac correction choices, scaffolds and biomaterials should contain various properties such as being biodegradable, biocompatible, flexible and durable and absence of immunogenicity and calcification. Due to the variety of sizes of patients' hearts and defects, designed scaffolds should have various sizes and be able to grow and adapt to the heart (*Schmidt & Baier, 2010*). The biomaterial should allow neo-vascularization for adequate oxygenation of the tissue, create minimal scarring tissue and thrombotic risk, as the latter could lead to life treatment with anticoagulants (*Witt et al., 2013*; *Miyagawa et al., 2011*). Furthermore, these biomaterials scaffolds should be bioactive, meaning they should enhance cellularization in vitro and in vivo, and optimize cell efficiency and degrade at a desirable rate (*Miyagawa et al., 2011*). What's more, biomaterials and scaffolds should be in abundance and cost-effective, as high cost could restrict development and use of it in TE as a routine therapeutic choice (*Avolio, Caputo & Madeddu, 2015*). The most common biomaterials for cardiac and vascular TE used today are synthetic, and natural polymers (*Files & Boucek, 2012*), and the electrospinning technique has been proven to be the most efficient way to produce scaffolds with these biomaterials (*Williams, 2004*).

### Synthetic polymers for cardiac scaffolds

The easiest way to have materials in abundance is to manufacture them. The need for suitable biomaterials for cardiac tissue repair has triggered development of synthetic polymers which are easy to fabricate and manipulate. These polymers can be manipulated with respect to their physical properties, molecular weight, heterogeneity index, and degradation speed (*Zhao et al., 2015*). Many synthetic polymers are biocompatible and have excellent mechanical properties which make them a popular choice for sutures and mesh production (*Files & Boucek, 2012*). Frequently used polymers in cardiac surgery are polyglycolic acid (PGA) and polylactic acid (PLA). These two polymers have been used as a single biomaterial or as 50:50 composite to reconstruct tissue-engineered vascular grafts for treating children with congenital heart disease (*Maeda et al., 2015*). *Carrier et al. (1999)* presented acceptable ultrastructural features and metabolic cell ability when cells were cultured on PGA scaffold in a rotating bioreactor (*Sugiura et al., 2018*). The rotating bioreactor increases cell adherence and decreases cell damage (*Sugiura et al., 2018*). The highest concern with synthetic polymers is their toxicity. Therefore, the use of poly-L-lactic acid (PLLA) has increasingly become more of interest. PLLA has demonstrated very good results when combined with bone marrow mesenchymal SC (BM-MSC) for vascular tissue engineering (*Masuda et al., 2008*). In vivo studies by Hashi et al. showed that nanofibrous scaffolds created with PLLA can remodel in both cellular and ECM content, similar to that of the native artery (*Masuda et al., 2008*). Both acellular and cellular scaffolds were implanted into the common carotid artery of live animal models (rats). The cellular scaffold was seeded with MSC and "exhibited very little platelet aggregation on the luminal

surfaces" compared to the acellular grafts. This is due to the antithrombogenic property of MSC (*Masuda et al., 2008*). PLLA, when degraded in the body, can "be excreted in carbon dioxide and water" (*Files & Boucek, 2012*). Polyurethane, unless copolymerized, is biocompatible but not biodegradable. Polyurethane has been successfully experimented in combination with other materials for cardiac tissue repair, such as siloxane films (*Hashi et al., 2007*; *Baheiraei et al., 2015*), cellulose (*Baheiraei et al., 2016*), urea (*Su et al., 2016*), PLLA (*Hernández-Córdova et al., 2016*). Poly (ε- caprolactone) in combination with other biomaterials have also been proven to be efficient composite for cardiac tissue repair. They have been used in combination with PLLA alone (*Tomecka et al., 2017*; *Sharifpanah et al., 2017*), PLLA and collagen (*Centola et al., 2010*), polypyrrole and gelatin (*Mukherjee et al., 2011*), polyglycolic acid (*Kai et al., 2011*), poly (hydroxymethyl glycolide) (*Sugiura et al., 2016*), chitosan and gelatin (*Castilho et al., 2017*).

Based on our understanding of the heart as an electroactive tissue, Hitscherich and colleagues have created a piezoelectric scaffold fabricated by electrospinning Polyvinylidene Fluoride-Tetrafluoroethylene (PVDF-TrFE) for cardiac tissue engineering (*Pok et al., 2013*). The combination of synthetic with natural polymers has been suggested to increase cell adherence. However, pure natural polymers have also been examined as an option for polymers (*Files & Boucek, 2012*).

### Natural polymers for cardiac scaffolds

Natural polymers are biodegradable, biocompatible and easily manipulated matrices composed of complex elements which make up the native tissue (*Hitscherich et al., 2016*). The natural polymers used so far for cardiac repair include collagen, gelatin, alginate, silk, fibrin, chitosan and hyaluronic acid (*Files & Boucek, 2012*). Despite their poor mechanical properties, they are good biomaterial for heart TE, as they have high biocompatibility, promote cell-binding and could biodegrade with no "additional treatment or modifications" (*Files & Boucek, 2012*).

Collagen is the most widely utilized natural polymer which is the most abundant ECM protein. It functions to guide biological processes, provide structural scaffolding, and tensile integrity (*Castilho et al., 2017*). Several kinds of literature have reported the use of various collagen types and their modifications in cardiac tissue repair (*Nelson et al., 2011*; *Herpel et al., 2006*; *Serpooshan et al., 2016*; *Dawson et al., 2011*; *Yu et al., 2017*; *Sun et al., 2017*; *Frederick et al., 2010*; *Hsieh et al., 2016*).

Fibrin can be manipulated to create gels, microbeads, and hydrogels (*Zhao et al., 2015*). Likewise, biological molecules like the growth factors can be incorporated (*Nie et al., 2010*). Fibrin glue can be used as a stand- alone therapy in cardiac tissue repair as it possesses intrinsic regenerative properties (*Menasché et al., 2014*). The success of fibrin patch seeded with human embryonic stem cell-derived cardiac progenitor cells (hESC-CPC) in non-human primate model (*Gil-Cayuela et al., 2016*) has resulted in its translation to the first case report of using hESC-CPC in severe heart failure with an encouraging patient functional outcome (*Menasché et al., 2015*). Other studies on fibrin have demonstrated its efficacy as a sealant after intramyocardial injection (*Terrovitis et al., 2009*); for myocardial tissue repair when seeded with adult stem cells, neonatal cardiac cell, and mesenchymal

stem cells (*Lluzià-Valldeperas et al., 2016*; *Tao et al., 2014*; *Ichihara et al., 2017*); to form aortic valves in tissue engineering (*Moreira et al., 2016*).

Chitosan has been experimented with and appears in the literature as a biomaterial for cardiac regeneration (*Camci-Unal et al., 2014*; *Wang et al., 2014*). Overall, in cardiomyogenesis, many researchers have agreed on the fact that chitosan seems to be more effective when combined with other factors enhancing integration of stem cells into cardiac tissues (*Chopra et al., 2006*).

Alginate, when used alone, has proven to have a remarkable effect on the function of heart models with myocardial infarction. Furthermore, seeding alginate with stem cells has proven to be more efficient in repair of the cardiac tissue (*Wang et al., 2012*; *Landa et al., 2008*; *Leor et al., 2009*; *Sabbah et al., 2013*).

The use of hyaluronic acid has been shown to be largely dependent on its molecular weight, and several kinds of literature have reported its successful use in cardiac tissue repair (*Bonafè et al., 2014*; *Yoon et al., 2009*; *Yang et al., 2010*; *Göv et al., 2016*).

Evidence of gelatin scaffold placed subcutaneously, and/or on infarcted myocardium in adult rat hearts have shown a good survival of the graft, vessel formation and junctions with recipient rat heart cells (*Li et al., 1999*). Gelatin was reported to sustain neonatal rat cardiomyocyte tissue in vitro for three weeks (*McCain et al., 2014*); supported the growth of human induced pluripotent stem cell (iPSC)-derived cardiomyocytes (*McCain et al., 2014*); its hydrogel seeded with autologous human cardiac-derived stem cell and basic fibroblast growth factor (bFGF) effectively released bFGF for repair of ischemic cardiomyopathy (*Yacoub & Terrovitis, 2013*); and several other studies have shown the efficacy of gelatin as a scaffold for cardiomyogenesis when seeded with cells (*Takehara et al., 2008*; *Navaei et al., 2016*; *Kudová et al., 2016*; *Cristallini et al., 2016*). Fibrinogen/Thrombin-based Collagen Fleece (TachoCombo) have been successfully used to secure hemostasis and enhance complete reconstruction of a large pulmonary artery defect in a canine model (*Okada et al., 1995*). Hence, this biomaterial may be used in reconstruction of the low-pressure pulmonary vessels during a cardiac surgery for a total anomalous pulmonary venous return or transposition of the great vessels.

Nevertheless, not all of these polymers can tune well for cardiac TE, and risk of inflammation still exists (*Sakai et al., 2001*).

### Native extracellular Matrix as scaffold

Native-specific ECM could be a category itself or part of natural polymers. ECM is collected from animal or donor tissue and processed for culturing cells (*Zimmermann, Melnychenko & Eschenhagen, 2004*). Studies have shown that the ratio of native ECM in culture could play a key role in stem cells (SCs) enhancement, differentiation, survival, and phenotype (*Duan et al., 2011*). Other studies have shown that contractible engineered heart patches cultured in ECM mixture and implanted in syngeneic Fischer 344 rats can vascularize, become innervated and survive up to 8 weeks in vivo (*Zimmermann, Melnychenko & Eschenhagen, 2004*).

Furthermore, ECM of decellularized and repopulated hearts and other organs are being used for drug testing (*Lewis, 2016*). An experiment on decellularized mouse hearts which

were repopulated with human cells through coronary vessels exhibited myocardium, vessel-like structures and intracellular Ca2+ transients contracted spontaneously and responded as expected to various drug interventions (*Lu et al., 2013*). It was concluded that heart "ECM could promote proliferation, specific cell differentiation and myofilament formation" (*Lu et al., 2013*). The option of ECM for TE could help overcome the challenges faced using synthetic and other natural biomaterials to replace tissue, valves or organs (*Iop et al., 2009*).

## Scaffoldless cell sheet

Another technique, which is independent of scaffolds, has been developed based on the cells' ability to connect via cell-to-cell junction proteins and create ECM (*Carrier et al., 1999*). The cells are cultured in normal conditions at 37 °C in a temperature-responsive polymer cultures dish. When the culture temperature conditions change, the cells detach from the polymer culture dish as one cell sheet (*Carrier et al., 1999*). This technique was developed to avoid inflammatory reactions and fibrotic deposits in the area of graft where scaffold was placed following degradation (*Carrier et al., 1999*). A study has shown that contractile chick cardiomyocyte sheets could function effectively around rat thoracic aorta when applied on host myocardium (*Witt et al., 2013*). This cell sheet could synchronize within 1 h of implantation with the host tissue (*Witt et al., 2013*). Similar results have been shown in 3D structures using several cell sheets aiming to create a thick cardiac patch (*Yamato & Okano, 2004*). The number of sheets is limited, as more than three exhibits poor vascularization. However, the combination of endothelial cells and cardiomyocytes is being examined to promote vascularization before implantation (*Carrier et al., 1999*). Table 3 summarizes the pros and cons of the various materials and biomaterials used in tissue engineering.

## Stem cells for tissue engineering

An equally important point in choosing a suitable biomaterial, scaffold or scaffoldless cell sheet, is the choice of the most appropriate cell types suitable for the TE. Stem cells (SCs) as a known cell source possesses the ability to differentiate toward Cardiac Muscle Cells/cardiomyocytes (CMCs), smooth muscle cells (SMCs) and endothelial cells (ECs), can regenerate cardiac tissue. Based on these properties, they play a key role in TE field. The currently used SCs in TE are summarized in Table 4.

## Embryonic stem cells

Embryonic stem cells (ESCs) are one of the cell sources which are used in TE approaches. ESCs are derived from the inner cell mass of preimplantation blastocyst (*Boroviak et al., 2014*). They can differentiate into all different cell types of three germ layers. Human ESC (hESC) could be a good candidate for cardiac tissue engineering. In a study conducted by Landry et al., hESC-derived cardiomyocytes (hESC-CMC) showed very good phenotype including myofibril alignment, density, morphology, contractile performance and gene expression profile which, however, was only confirmed after 80–120 days in vitro culture (*Lundy et al., 2013*). Various groups apart from Landry and colleagues conducted studies to confirm the successful differentiation of ESC to cardiomyocytes as presented in a review by Boheler and colleagues in 2002 (*Boheler et al., 2002*). *Duan et al.*

**Table 3** Advantages and disadvantages of materials and biomaterials used in TE.

|  | Artificial prosthesis | Biological prosthesis | Biomaterial scaffolds | Scaffoldless tissue |
|---|---|---|---|---|
| **Advantages** | • Available in abundance<br>• Many different sizes<br>• Long term results available | • Available in abundance<br>• No requirement for life-long treatment with anticoagulants | • Good mechanical properties<br>• Ultrastructural features<br>• Cell adherence<br>• Biocompatibility<br>• Biodegradable<br>• 3D-priting allows any shape and size | • No need for scaffold<br>• Spontaneous and synchronous pulsation<br>• Could create tubular construct<br>• Can grow with host |
| **Disadvantages** | • Impact & friction wear<br>• Inflammation<br>• Calcification of valve<br>• Less flexible than natural tissue<br>• Life-long treatment with anticoagulants<br>• Do not grow with the patients' heart | • Risk of cytotoxicity<br>• Inflammation<br>• Loss of mechanical properties<br>• Calcification in infants and children<br>• Immunological reactions | • Some present toxicity<br>• Risk of inflammation<br>• Not all tune well with the heart | • Limited number of cell sheets (max 3)<br>• Poor vascularization in more than 3 cell-sheets |

**Table 4  Scaffolds and SCs used for TE in some study models.**

| Engineered tissue | Scaffold | Type of SCs | Study models | Reference |
|---|---|---|---|---|
| Heart valve | Synthetic biodegradable non-woven PGA mesh | Human Chorionic villi-derived cells & hCB- EPCs | Culture in bioreactor | 164 |
| | Synthetic biodegradable | hAFSCs | Culture in bioreactor | 166 |
| | Porcine decellularized scaffold | BM-MSCs & BM-MSCs | Lambs | 176 |
| Vascular graft | Various synthetic biodegradable | Human Umbilical CB-EPCs | Static conditions & biomimetic flow system | 165 |
| | Biodegradable non-woven PGA | BM-MNCs | Mice | 174 |
| | Biodegradable PLA & PGA | BM-MNCs | Human | 79 |

*(2011)* investigated how native cardiac ECM could affect hESC differentiation. This group processed porcine hearts to collect digested cardiac ECM which then mixed with collagen to create a hydrogel for cell cultivation purposes. The cultured hESCs on biomaterials comprised of 75% native cardiac porcine ECM and 25% hydrogel with no additional growth factors have shown a great differentiation with cardiac troponin T expression and contractile behavior, compared to the hydrogel with a smaller ratio of native ECM. Based on various studies, ESCs could be a good option for cardiac TE (*Zimmermann, Melnychenko & Eschenhagen, 2004*). Additionally, some factors such as ethical concerns, provoked immunogenicity, and risk of tumorigenesis make the ESCs a very controversial choice of cell source for TE (*Schmidt & Baier, 2010*).

## Induced pluripotent stem cells

Another type of SCs which are used in TE is induced pluripotent stem cells (iPSCs). The iPSCs are somatic cells which are reprogrammed to behave like ESC and show the same properties. The iPSCs can differentiate into all three germ layers (*Takahashi et al., 2007*; *Yu et al., 2007*). Takahashi and his group was the first group who was able to reprogram the somatic adult cells like fibroblasts to iPSCs using viral vectors to introduce four key factors OCT4, SOX2, c-Myc, and KFL-4 to fibroblasts (*Takahashi et al., 2007*). This method was used to reprogram fibroblasts to embryonic-like cells and from this state differentiate them into a relevant type of cells (*Takahashi et al., 2007*). Ludry et al. have also shown that human iPSC-derived cardiomyocytes (hiPSC-CMCs) present the same characteristics as hESC-derived cardiomyocytes in long-term in vitro culture (*Lundy et al., 2013*). Lu and colleagues presented the successful repopulation of decellularized cadaveric mouse heart with hiPSC-derived multipotential cardiovascular progenitors (*Lundy et al., 2013*). They also demonstrate that the heart ECM promotes proliferation, differentiation and myofilament formation of CMs from the repopulated hiPSC-derived cells. Furthermore, they have checked the electrical coupling of these cells and also examined the constructive ability of repopulated heart using electrocardiogram which presented arrhythmia. Lu et al. have further examined the effects of pharmacological agents on repopulated heart and observed remarkable responses (*Zimmermann, Melnychenko & Eschenhagen,*

*2004*). This model is explored as an option to personalized medicine concerning drug testing/discovery (*Zimmermann, Melnychenko & Eschenhagen, 2004*; *Bosman et al., 2015*). Specifically, for CHD which presents such a variety of profiles, individual patient-specific human models' development could help to understand how each patient would respond to existing pharmaceutical treatments. Even though it may not be possible to create actual organ heart models with individual clinical features of the disease (*Caputo et al., 2015*).

Nonetheless, like ESC, iPSC has demonstrated tumorigenesis (*Schmidt & Baier, 2010*). Ieda and colleagues showed a direct transdifferentiation of fibroblasts to functional cardiomyocytes using three key factors, Gata4, Mef2c, and Tbx5, within a very short time and suggested that direct reprogramming could reduce the risk of tumorigenesis (*Ieda et al., 2010*). Still, using viral vectors for reprogramming procedure is problematic and involves various risks (*Schmidt & Baier, 2010*). Therefore, today more different ways for iPSC production are being used and investigated to find out safer and more effective alternatives for this reprogramming procedure (*Tarui, Sano & Oh, 2014*; *Lüningschrör et al., 2013*). In the event this problem is solved, iPSC could be the safest type of cell sources for TE as they will not provoke any immune-response and cell harvesting procedure to produce iPSCs is not life-threatening for the patients. Moreover, iPSC raises less ethical concerns in comparison to ESC or fetal SC.

## Prenatal, perinatal, and postnatal stem cells

Prenatal, perinatal and postnatal SCs are other cell sources used in TE, which include chorionic villi derived multipotent SCs, amniotic fluid-derived SCs (AFSCs), umbilical cord blood derived-endothelial progenitor cells (UCB-EPCs) and umbilical cord- or cord blood-derived- multipotent SCs (*Webera, Zeisbergera & Hoerstrup, 2011*). UCB progenitors, like endothelial progenitor cells (EPCs), have distinctive proliferative properties in comparison to other cells sources (*Murohara et al., 2000*). This category of SCs is exceptionally important as the child's own SCs could be used for heart TE, for CHD patients who are diagnosed before birth. Immunogenicity or an additional procedure to harvest autologous SCs from the infant or child could be avoided in this procedure. Furthermore, it has been proven that AFSCs do not form teratomas in contrast to ESCs and iPSCs (*De Coppi et al., 2007*). All categories of these cells have been investigated with remarkable results on engineered valves and vascular grafts (*Webera, Zeisbergera & Hoerstrup, 2011*; *Schmidt et al., 2006*; *Schmidt et al., 2004*; *Schmidt et al., 2007*; *Yao, 2016*; *Petsche Connell et al., 2013*). This type of SCs is not applicable to adults who have been diagnosed later in life with CHD.

## Adult stem cells
### Bone marrow-derived stem cells

Apart from UCB, EPCs can be found in the peripheral blood (PB-EPCs) and bone marrow (BM- EPCs) of adults (*Asahara et al., 1997*; *Zisch, 2004*). However, bone marrow is a richer source of EPCs in comparison to peripheral blood. *Asahara et al. (1997)* identified the CD34+ mononuclear hematopoietic progenitor cells in the peripheral blood which in vitro presented endothelial-like characteristics. The EPCs which are originated from BM considered to play a crucial role in endothelial repair, and they have been suggested for treatment of ischemia patients and vascular TE with very encouraging results (*De Coppi*

*et al., 2007*; *Schmidt et al., 2004*; *Zisch, 2004*; *Olausson et al., 2014*). A successful complete endothelium regeneration of decellularized canine carotid arteries has been reported in animal studies using PB-EPCs (*Zhou et al., 2012*). In addition to vascular TE, the EPCs have been assessed for tissue-engineered heart valves (*Sales et al., 2011*).

Bone marrow derived EPCs are greatly involved in de novo vessel formation and neovascularization in pathological conditions like ischemia and cancer (*Asahara et al., 1997*). Similar applications to PB-EPCs and prenatal EPCs have been recorded for vascular graft in the congenital heart surgery using bone marrow-derived stem cells (*Mirensky et al., 2010*). Mirensky and colleagues used sheets of non-woven PGA mesh as a scaffold to create vascular graft in combination with human bone marrow mononuclear cells (BM-MNCs). The results were very encouraging as no aneurysm or thrombotic incidence were reported, despite the absence of anticoagulants. This group suggests this method as a suitable vascular treatment for CHD based on their results from 6 weeks follow up after graft implantation, which has shown signs of degradation, and it was fully accommodated by the host's cells (*Mirensky et al., 2010*). Nonetheless, the host mice were at full-growth which makes it questionable how successful this application could be in a developing animal model. Interestingly only one week after implantation no human BM-MNCs were detected, which suggests that BM-MNCs play a paracrine role rather than cell replacement (*Zisch, 2004*). More studies have reported similar results with the same conclusion (*Schmidt & Baier, 2010*; *Fernandes et al., 2015*). Furthermore, an investigation on 25 young patients under 30 years old who underwent extracardiac total cardiopulmonary connection with BM-MNCs engineered vascular graft has also presented convincing results. A long-term patient follow-up has shown zero deaths in relation to the implanted grafts, no thromboembolic, hemorrhagic, or infectious complications, however, 6 of them developed grafts stenosis which was treated successfully (*Schmidt & Baier, 2010*). Despite the encouraging results with BM-MNCs, BM-MSCs still present more advantages. These advantages are, for example, their ability to differentiate into a variety of cell types even progenitor cells; relatively easy procedure for their collection, isolation, storage, and proliferation; presenting a similar phenotype to the valve cells; present anti-thrombogenic properties; and their immunogenicity is manageable (*Masuda et al., 2008*; *Iop et al., 2009*). In a comparative study, *Vincentelli et al. (2007)* examined short- and long-term characteristics of the porcine decellularized scaffold which were processed with in-situ injections of BM-MNCs and BM-MSCs, before transplanting in a lamb of animal models. Short-term results did not show any significant differences. However, the 4 months (long-term) follow up has shown a significant decrease of transvalvular and distal gradients, more inflammatory reaction, more structural deterioration as well as calcification, and a thick fibrous pannus around the suture line in the BM-MNCs group. These observations in the BM-MNCs group were significantly different from the BM-MSCs group (*Vincentelli et al., 2007*).

### Cardiac progenitor cells

Cardiac progenitor cells (CPCs) or also known as Cardiac Stem Cells (CSCs) are a type of cells which are found in the adult heart and express stem cell factor receptor kinase c-kit+,
also shown to be negative for the markers of blood/endothelial cell lineage (*Beltrami et al., 2003*). CPCs have only been in the spotlight for about 15 years now with the credit given to Beltrami and colleagues as they demonstrated the self-renewing and multipotent characteristics of these cells; differentiating into all three different cardiogenic cell types which are cardiomyocytes, smooth muscle cells and endothelial cells (*Beltrami et al., 2003*). More recently, a study carried out by Vicinanza and colleagues helped to lay to rest the controversies about c-kit+ cardiac cells by demonstrating that only a small fraction of the c-kit+ adult cardiac stem cells possess the tissue-specific progenitor properties (*Vicinanza et al., 2017*). The study was carried out in adult Wistar rats in which an experimental acute myocardial infarction (MI) was induced and the border of the infarcts were injected with GFP+c-kit+ for the first group, "GFP-expressing CD45-c-kit+ CSCs (CSC$^{GFP}$)" for the second group, and the placebo group were injected with PBS. The study showed that CSC$^{GFP}$ only significantly reduced apoptosis and hypertrophy of the cardiomyocytes, it also significantly reduced scarring and improved ventricular functions (*Vicinanza et al., 2017*). The study further demonstrated that about 10% of the overall c-kit+ cardiac cells are CD45-c-kit+ , and only about 10% of these are clonogenic and multipotent. Therefore, it was inferred that only about 1–2% of the total c-kit+ myocardial cells have a demonstrable multipotent CSC phenotype (*Vicinanza et al., 2017*). These c-kit+ CD45-(deletion of which also renders the cells CD31-, CD34-) CPCs were also shown to express to some degree Sca-1, Abcg2, CD105, CD166, PDGFR- $\alpha$, Flk-1, ROR2, CD13, and CD90 (*Vicinanza et al., 2017*). The number of CPCs have however been shown to be significantly higher in neonates but dramatically decreases after the age of 2 (*Chaikof, 2007*; *Bosman et al., 2011*). Thus, these CPCs in combination with a suitable scaffold, could be the answer to treat a number of CHD, as CPCs could be collected during palliative surgery or via endomyocardial biopsy (*Saxena, 2010*; *Torok, 2015*).

The potential SCs and biomaterials for TE in CHDs are represented in the Fig. 1 below. Also, Fig. 2 below shows the promising strategies for the treatment of CHDs.

## STANDPOINT

Due to the high number of patients as well as newborns who are suffering from CHD and also high costs of their implications, it is necessary and vital to making intensive research for finding an effective treatment for CHD patients. Stem cell research has shown remarkable results in all kinds of tissue engineering including skin (*Alessandri, Emanueli & Madedu, 2004*), cartilage (*Metcalfe & Ferguson, 2007*), vascular (*Makris et al., 2015*), ocular (*Chlupác, Filová & Bacáková, 2009*) and cardiac tissues (*Miyagawa et al., 2011*; *Sugiura et al., 2018*; *Karamichos, 2015*; *Kane et al., 2014*).

The importance of engineered cardiac tissue lies in the fact that synthetic non-degradable materials cannot adjust to the patient's developing body. A number of patients who suffer from CHD are adults, and they are more suitable for this type of therapy. However, many patients with severe CHD are infants and children whose bodies's are constantly developing. Although various synthetic and natural biodegradable biomaterials have been used so far which have shown good results, the one with the best degradation rate is yet to be found.

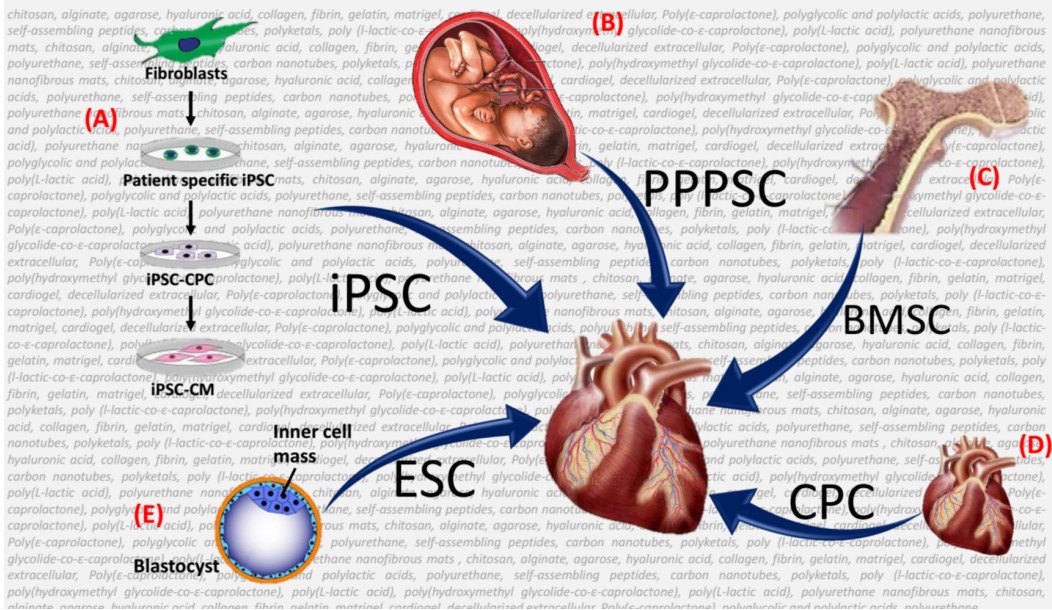

**Figure 1** **Schematic of the different types of stems that can be used on the biomaterial backbone for cardiovascular tissue engineering (TE).** This schema represents the different types of stem cells that can be used on the biomaterial backbone (depicted as the background characters) for cardiovascular Tissue Engineering (TE). (A) Induced pluripotent stem cells (iPSCs) derived from fibroblast. (B) Prenatal, Perinatal, and Postnatal Stem cells (PPPSCs) are derived from amniotic fluid, umbilical cord, and chorionic villi. (C) Bone Marrow Stem Cells (BMSCs) such endothelial progenitor cells (EPCs) and mesenchymal stem cells (MSCs) can easily be isolated from the bone marrow. (D) Cardiac progenitor cells (CPCs) can be harvested during palliative surgery or endomyocardial biopsy. (E) Embryonic stem cells (ESCs) derived from the inner cell mass of the blastocyst.

There are various complications related to existing surgical treatments and scaffolds which cannot be ignored. Calcification, inflammatory reaction and life-long anticoagulants treatment are the most important known complications for the conventional methods of CHD treatment (*Zilla et al., 2008*; *Brown, 2005*). The complexity of CHD makes TE possibly the most suitable solution for treatment of patients with CHD.

## CONCLUSION

The replacement or correction of a malformation in a complex system like the cardiovascular system could only be successful with tissues which can mimic the native heart and vascular tissues. SCs have opened the door to such treatments. The best SC candidates and biomaterials are yet to be identified, despite the encouraging results. All different types of SCs which have been investigated so far still present some disadvantages. Extensive research would be required to enable deeper understanding, solve drawbacks, and promote SCs use for tissue engineering in the future. All the efforts channeled at obtaining proper legal regulation for using SCs, developing new technologies for scaffold production as well as scaffoldless techniques, developing faster and safer methods for

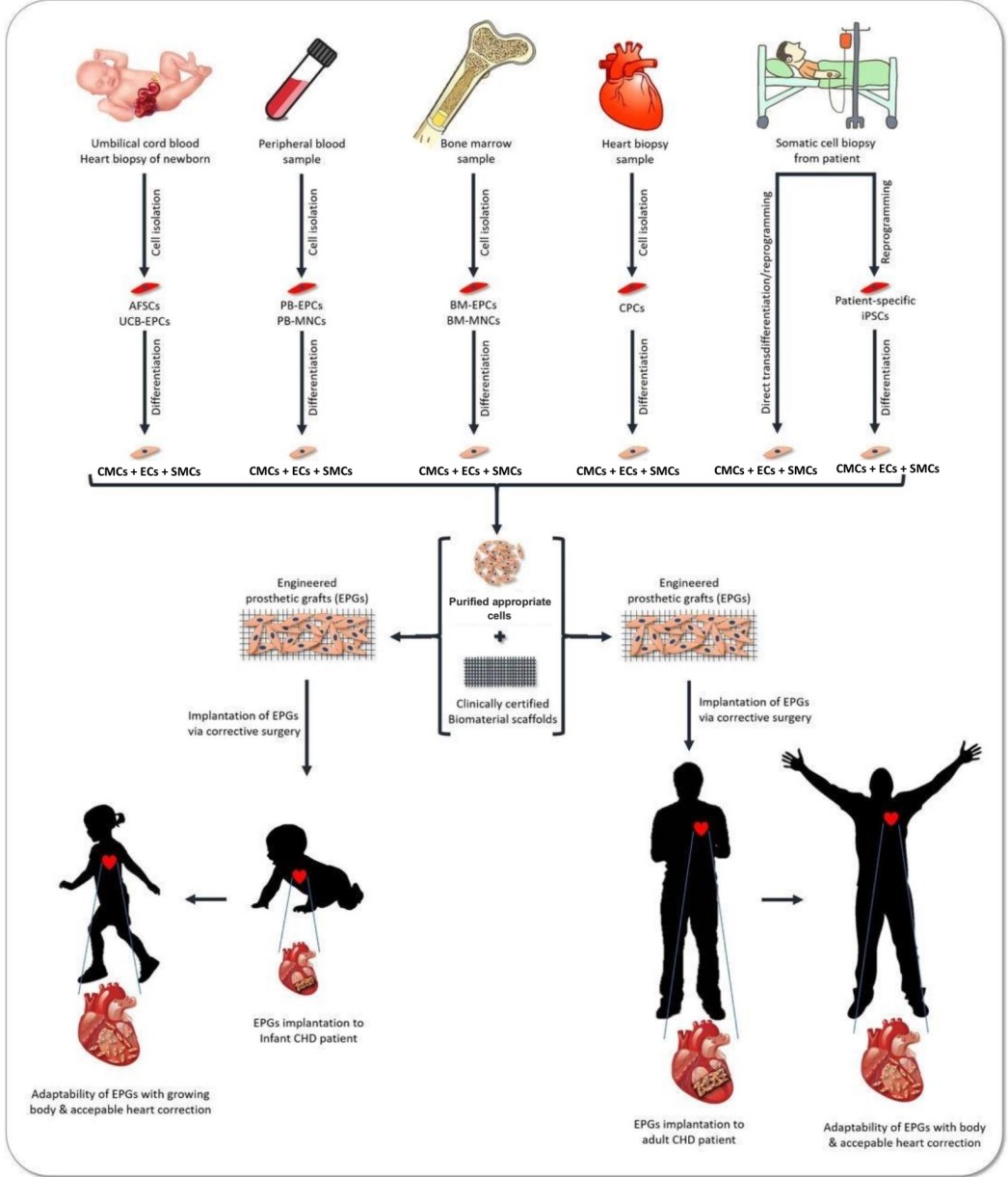

**Figure 2** **Promising strategies for CHDs treatment.** The schematic diagram represents the potential of stem cells (SCs) and tissue engineering (TE) for corrective surgical treatment of infants as well as adolescent patients with Congenital Heart Disease (CHD). Various sources for stem cells (SCs) are presented here as alternatives to harvesting the appropriate stem cells (SCs) which can be used to seed on clinically certified biomaterial scaffolds for reconstructing functional cardiac tissue-engineered grafts. These grafts could be implanted via the corrective surgery into the heart of infants and adolescent patients with congenital heart disease (CHD) for definitive correction of cardiac defects. These optimized cardiac-tissue engineered grafts should have the potential to grow in parallel with the child, while lacking any tumorigenicity, immunogenicity, thrombogenicity, calcification, or other risk factors.

producing patient-specific iPSCs, and research into the effectiveness of SCs in TE for treatment of CHD, predicts a very positive future for patients, researchers and surgeons.

## ACKNOWLEDGEMENTS

The authors wish to appreciate Professor Costanza Emanueli, Professor of Cardiovascular Medicine, School of Clinical Sciences, University of Bristol and the Bristol Heart Institute, and Professor Massimo Caputo, Professor of Congenital Heart Disease and Consultant in Cardiothoracic Surgery, University of the Bristol and the Bristol Heart Institute, for assigning this work. The authors also wish to appreciate the contributions of Precious Anthony, Favour Anthony, and David Otohinoyi at the All Saints University School of Medicine.

### Funding
The authors received no funding for this work.

### Competing Interests
The authors declare there are no competing interests. Antonia Mantakaki is an employee of Teleflex Incorporated.

### Author Contributions
- Antonia Mantakaki, Adegbenro Omotuyi John Fakoya and Fatemeh Sharifpanah prepared figures and/or tables, authored or reviewed drafts of the paper, approved the final draft.

### Data Availability
The research in this article did not generate any data or code (this work is a literature review).

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
