# Peer review of "Recent advances and challenges on application of tissue engineering for treatment of congenital heart disease"

_PeerJ, doi:10.7717/peerj.5805_

## Round 0.1 · original submission · Major Revisions

Dear Antonia and Adegbenro,

Thank you for your submission to PeerJ. I am in agreement with the reviewers that this review is cross disciplinary and timely. The content is of interest as new treatments for CHD are highly sought after and we are seeing the application of cellular therapies to treat CHD in the clinical setting.

The use of TE approaches to improve outcome of cellular therapies is also of interest not just to the cardiac biology field but across the multi disciplinary field of regenerative medicine.

I do however agree with the reviews that the changes they mentioned are need to make this review more complete. I would like to invite you to address the concerns raised.

Regards
Annette Meeson

·

Basic reporting

This is a nice review article on the treatment of congenital heart disease using tissue engineering approaches, including describing the different types of stem cells that could be used.

I have the following suggestions to improve the article:

1. Line 83, the authors state that 'Several materials and biomaterials have been used in the surgical management of CHDs...' Could a couple of examples be described?
2. Line 86, write Tissue Engineering in full the first time it is mentioned
3. Line 101, 'This evidence shows the existence of Lin c-kit+ cells which confirm cardiomyocyte proliferation and cardiac regeneration'. Considering the confusion over the phenotype of a cardiac stem/progenitor cell, I suggest to re-write this sentence as 'This evidence shows the existence of cardiac stem/progenitor cells which can differentiate into new cardiomyocytes and participate in cardiac regeneration'.
4. Line 102, Could the authors give the original reference (Porrello et al. 2011, Science) and describe in more detail that the neonatal 1 day old Mouse heart can regenerate after partial surgical resection, but this capacity is lost by 7 days of age, and in adulthood.
5. Line 134, define Cyanotic and Acyanotic
6. Line 125, 3.0 Congenital Heart Disease: Types, malformations and surgical intervention. It would be good if a Table could be inserted that summarises the types of CHD, and their manifestations.
7. Line 170. Reword the sentence beginning 'The main challenge....' to 'The main challenge with artificial materials is alterations in the size and function of the heart from neonate, infant to adulthood, which current artificial material cannot adjust too.'
8. Line 225, provide clarity on '...about the actual function of cells following citation on the clot'.
9. Line 309-311, explain why PLLA combined with BM-MSC has demonstrated good results for vascular tissue engineering, and in which context - in vitro or in vivo, and which type of CHD?
10. Line 372, write in full SC first time it is written.
11. Line 373-374, describe in more detail the results of study reference 110. Which in vivo disease model was this tested in?
12. Line 436, change great to good
13. Figure 1 shows the different stem cells and their differentiation into cardiomyocytes, however what about endothelial cell and smooth muscle cell differentiation? And also this cartoon of differentiation is not reflective of the text. More literature needs to be described on ESCs, iPSCs, UCB, AFSCs, BM-MNCs or MSCs, CPCs ability to differentiate into cardiomyocytes in vitro and their cardiac regenerative potential in in vivo animal models, and preferably CHD models.
14. Please state for each study described whether the work was carried out in vitro or in vivo
15. Line 538, state the phenotype of CPCs and the markers they express as well as c-kit, which are Sca-1+, PDGFr-alpha+, CD31-, CD34-, CD45-, tryptase-. See and reference Vicinanza et al. 2017, Cell Death & Differentiation.
16. Line 538, change 'proliferating' to 'self-renewing'
17. Line 539, re-write 'characteristics; differentiating into all three cardiogenic cells types, which are cardiomyocytes, smooth muscle cells and endothelial cells'. Note the change of epithelial cells to endothelial cells.

18. Line 74, CDH should be CHD
19. Line 202, insert 'the' between repairing and heart.
20. Line 203, change have to has
21. Line 210, change was to is
22. Line 236, change by to from
23. Line 252, change contacted to carried out
24. Line 283, remove s
25. Line 353, change myocardial to myocardium
26. Line 362, should hemostasis be homeostasis
27. Line 416, 'write An equally important point in choosing a suitable biomaterial....
28. Line 418, abbreviate cardiomyocyte to CM not CMC
29. Line 468, change 'were able to make a' to 'showed'
30. Line 485, change form to from
31. Line 526 change 'in lamb of animal models' to 'in a lamb animal model'.
32. Line 557, Figure 1 legend, change will to could
33. Line 558, Figure 1 legend, change This to These
34. Line 559, Figure 1 legend, remove 'a' and 'growth'
35. Line 569, change lays to lies
36. Line 573, remove 'with'
37. Line 583, remove 's' from where is written 'the best SCs candidates..'
38. Line 585, insert the word 'enable' between 'to' and 'deeper'
39. Line 586, change understand to understanding
40. Define abbreviations used Figure 1 in legend

Experimental design

Not applicable as it's a review

Validity of the findings

Not applicable as it's a review

Reviewer 2 ·

Basic reporting

This review article is timely, and of broad and cross-disciplinary interest. It is written clearly and unambiguously, using a professional English mostly throughout (some minor phrase structure revisions are advised). In addition, the manuscript's structure should be revised to comply with the publication's format (add "Survey Methodology" section, use the correct section numbers).
However, in its substance, this manuscript does not represent a meaningful addition to other relatively recent literature reviews, which have approached the exact some subject very thoroughly and with a clear point of view.

Experimental design

no comments

Validity of the findings

As this literature review is limited to briefly state the recent advances on the application of stem cells and tissue engineering approaches towards CHD treatment, with minimal critical analysis of the previous art, it is difficult to impress the reader the huge potential (and corresponding challenges) of the field, now and in the future.

Additional comments

In the manuscript entitled 'Stem cells, tissue engineering and congenital heart disease', the authors aim at providing an introduction to the pathology followed by a throughout overview of the many approaches using stem cells and tissue engineering methods to develop feasible therapies. The authors provided an ample array of such examples, however there is an inherent lack of structure to this review that hampers its potential impact. Importantly, the authors tend to dump information with little historical or methodological context, making very difficult to the reader to understand why a particular approach was developed and used, what are its strengths and weaknesses, and where to go from there (i.e. what possible future it might have in the clinic).

Moreover, the authors tend to exercise only a minimal critical analysis of the many studies they refer, going so far as to cite verbatim without further insights (e.g. '...It was concluded that heart “ECM could promote proliferation, specific cell differentiation and myofilament formation” [113].') A more carefully prepared review should have a personal point of view to highlight, using the available literature to support their position and guide the reader; as it is, this review is limited to be just an enumeration of previous studies.

Fortunately, these issues can be addressed by improving the structure of the manuscript, e.g. by presenting the work in an historical context, and/or by analyzing the described advances/innovations more critically, e.g. to highlight those showing better outcomes/higher accessibility to the clinic/more robust and diversified applications/better promise considering their specific challenges and limitations. The addition of a table for 'Frequent types of CHDs, their main challenges and corresponding therapeutic approaches so far', as well as a revised table 2 compiling a more exhaustive list of examples, grouped by type of engineered tissue or by type of biomaterial or stem cells used.

---

## Round 0.2 · Minor Revisions

Please take a look and add to your manuscript comments on the Vicinanza et al paper of 2017 this is relevant to the subject of your review

Regards
Annette Meeson

·

Basic reporting

NA

Experimental design

NA

Validity of the findings

The authors have responded to my comments and made changes, except 1.

When the authors describe the Cardiac progenitor cells starting line 517, the CPCs are CD34-negative, tryptase-negative (please correct, line 519), and they should reference the paper by Vicinanza et al. 2017, Cell Death & Differentiation, which is the most up-to-date description and characteristics of the CPCs. The authors should also describe the in vivo experiments in Vicinanza et al. 2017, Cell Death & Differentiation which show the regenerative potential of clonogeneic CSCs following transplantation after MI.

---

## Round 0.3 · accepted · Accept

Thank you for your revision. I am happy to Accept the paper.

·

Basic reporting

I am happy with the revisions.

Experimental design

n/a

Validity of the findings

n/a

Additional comments

n/a